# Expanding radiogenic strontium isotope baseline data for central Mexican paleomobility studies

Sofía I. Pacheco-Forés[1]*, Gwyneth W. Gordon[2], Kelly J. Knudson[1]

**1** Center for Bioarchaelogical Research, School of Human Evolution and Social Change, Arizona State University, Tempe, AZ, United States of America, **2** School of Earth and Space Exploration, Arizona State University, Tempe, AZ, United States of America

* sipachec@asu.edu

**Data Availability Statement:** All relevant data are within the paper and its Supporting Information files.

**Funding:** Fieldwork and laboratory analyses were supported by National Science Foundation (https://

## Abstract

Radiogenic strontium isotopes ($^{87}$Sr/$^{86}$Sr) have long been used in analyses of paleomobility within Mesoamerica. While considerable effort has been expended developing $^{87}$Sr/$^{86}$Sr baseline values across the Maya region, work in central Mexico is primarily focused on the Classic period urban center of Teotihuacan. This study adds to this important dataset by presenting bioavailable $^{87}$Sr/$^{86}$Sr values across central Mexico focusing on the Basin of Mexico. This study therefore serves to expand the utility of strontium isotopes across a wider geographic region. A total of 63 plant and water samples were collected from 13 central Mexican sites and analyzed for $^{87}$Sr/$^{86}$Sr on a Thermo-Finnigan Neptune multi-collector inductively coupled plasma mass spectrometer (MC-ICP-MS). These data were analyzed alongside 16 published $^{87}$Sr/$^{86}$Sr values from two additional sites within the region of interest. A five-cluster k-means model was then generated to determine which regions of the Basin of Mexico and greater central Mexico can and cannot be distinguished isotopically using $^{87}$Sr/$^{86}$Sr values. Although the two clusters falling within the Basin of Mexico overlap in their local $^{87}$Sr/$^{86}$Sr ranges, many locations within the Basin are distinguishable using $^{87}$Sr/$^{86}$Sr values at the site-level. This study contributes to paleomobility studies within central Mexico by expanding knowledge of strontium isotope variability within the region, ultimately allowing researchers to detect intra-regional residential mobility and gain a greater understanding of the sociopolitical interactions between the Basin of Mexico and supporting outlying regions of central Mexico.

## Introduction

Researchers have long debated the importance of migration in the cultural development of central Mexico. A number of archaeological [1–4], morphological [5–9], and genetic [10–11], analyses indicate that the Basin of Mexico attracted multiple waves of migrants from across greater Mesoamerica throughout pre-Hispanic times. Biogeochemical studies of radiogenic strontium ($^{87}$Sr/$^{86}$Sr) isotopes have proven effective in directly testing the presence of migrants

www.nsf.gov) grants 2013155229 and 1744335 awarded to SIPF as well as a School of Human Evolution and Social Change (https://shesc.asu. edu) grant awarded to SIPF. The funders had no role in study design, data collection and analysiss, decision to publish, or preparation of the manuscript.

**Competing interests:** The authors have declared that no competing interests exist.

within the Basin, particularly at the Classic period city of Teotihuacan [12–17]. While determining "local" ranges of variation in $^{87}$Sr/$^{86}$Sr values is essential for the further application of this method, central Mexican radiogenic strontium data outside of Teotihuacan remain limited. Price and colleagues [18] established a regional expected $^{87}$Sr/$^{86}$Sr range for the Basin of Mexico as a whole, but no studies examine $^{87}$Sr/$^{86}$Sr variability within the Basin or central Mexico.

This study investigates radiogenic strontium variability within the Basin of Mexico and greater central Mexico, further facilitating paleomobility studies within the region. We first discuss the use of strontium isotopes in paleomobility within Mesoamerica and beyond and then consider geologic expectations for $^{87}$Sr/$^{86}$Sr values within central Mexico and the Basin of Mexico. Finally, we present biogeochemical data on modern plant and water samples ($n$ = 63), analyzing them alongside published data ($n$ = 16) [12,15] to characterize biogeochemically distinguishable zones within the Basin of Mexico and central Mexico.

## Strontium isotopes in studies of paleomobility

Radiogenic strontium isotopes are one of several isotopic systems that have been used to characterize paleomobility [19–23]. $^{87}$Sr/$^{86}$Sr values reflect regional geologic variability [24]. Biologically available strontium present in soil and groundwater is incorporated into local plants and subsequently into hydroxyapatite, the hard tissues (including bone and enamel) of animals ingesting that vegetation [25–28]. By comparing the strontium isotopic values in human and animal hard tissues mineralizing at different times over the life course, bioarchaeologists can reconstruct prehistoric patterns of mobility between distinct geologic zones over the life course [12,20,29–31].

## Strontium isotope systematics

Strontium is an alkaline earth metal typically found in rock, water, soil, plants, and animals at the parts-per-million (ppm) level [24,32]. Of the four naturally occurring strontium isotopes, $^{87}$Sr is radiogenic and is produced by the slow radioactive decay of rubidium ($^{87}$Rb). Thus, the abundance of $^{87}$Sr in a given region varies by the age and composition of local bedrock minerals [24,32]. Geologically older igneous and granitic formations rich in parent $^{87}$Rb are enriched in $^{87}$Sr ($^{87}$Sr/$^{86}$Sr > 0.750) compared to geologically younger volcanic basalts, rhyolites, or andesites ($^{87}$Sr/$^{86}$Sr ≈ 0.702–0.704), while marine carbonates and metamorphic formations often have intermediate values [24,33,34]. There is a large range of variation in comparison to the instrumental error of mass spectrometer measurements, which can generate accurate measurements up to the fourth decimal place or better (± 0.00001) [31,34]. As such, geologic maps of bedrock types and ages can be used to predict expected $^{87}$Sr/$^{86}$Sr variation.

Predictions based solely on geologic maps of bedrock types, however, are not always accurate. A number of factors, including the modification of source rock by erosion and preferential weathering of mineral with more radiogenic signatures, the addition of material from wind-derived material, and sea spray, can be mixed to produce different bioavailable strontium ratios that ultimately end up incorporated in hydroxyapatite [34–36]. Thus, researchers have undertaken strontium isotope studies of local water sources, soils, plants, and animal bones to more accurately characterize bioavailable strontium variability in a given environment [28,37–43].

## Strontium isotopes and paleomobility across Mesoamerica

Studies using $^{87}$Sr/$^{86}$Sr isotopes to reconstruct paleomobility throughout Mesoamerica have increased dramatically in recent years as archaeologists seek to directly test models of ancient

migration, diaspora, and mobility within the region [44]. Researchers have used radiogenic strontium isotopes to reconstruct ancient migration patterns [12,17,41,45–50], the geographic origins of sacrificial victims [13,51], and past animal trade and management networks [52,53], as well as long distance material culture trade networks [54] and historic diasporas [55,56].

Other studies have focused on characterizing $^{87}Sr/^{86}Sr$ variability across Mesoamerica. Hodell and colleagues [40] carried out an extensive study of radiogenic strontium variability across the Maya region of southern Mexico, Belize, and Guatemala to identify isotopically distinct sub-regions. Similarly, Price and colleagues [18] analyzed $^{87}Sr/^{86}Sr$ ranges more broadly across Mesoamerica. While they report a local range of $^{87}Sr/^{86}Sr = 0.7046–0.7051$ for the Basin of Mexico, little published data exist examining variability within central Mexico and the Basin itself.

## Central Mexican geography, geology, and geochemistry

Understanding regional geology is essential to the study of variability in radiogenic strontium isotope values within central Mexico, which is defined here as including the modern Mexican states of Mexico State, Hidalgo, Puebla, Tlaxcala, and Morelos, as well as Mexico City. Geologists have divided Mexico into several geologically and physiographically distinct morphotectonic provinces (Fig 1). However, only three morphotectonic provinces—the Sierra Madre Oriental, Mexican Volcanic Belt, and Sierra Madre Sur—make up central Mexico.

The geology of central Mexico is a complex mixture of recent volcanic highlands and older marine sedimentary deposits, along with a variety of metamorphic rocks [59–61]. The northern portion of central Mexico is comprised of the Sierra Madre Oriental mountain range. The Sierra Madre Oriental is primarily made up of orogenic Mesozoic Jurassic and Cretaceous sedimentary carbonates, sandstones, and shales of marine origin with some metamorphic Precambrian and Paleozoic gneiss and schist outcrops [61,62]. Immediately to the south and forming the heart of central Mexico is the Mexican Volcanic Belt, which extends from the Pacific to Gulf coasts. The Mexican Volcanic Belt is a Cenozoic volcanic plateau with central basaltic andesites forming during the late Miocene and early Pliocene and younger southern andesites, dacites, and rhyolites forming more recently during the Quaternary [59,61,63–65].

Finally, the southern edge of central Mexico is defined by the Sierra Madre del Sur mountain range. The Sierra Madre del Sur is the most geologically complex morphotectonic province in Mexico, composed of a northern segment of Mesozoic Jurassic and Cretaceous sediments and volcanic rock outcrops partially covered by Cenozoic volcanic and sedimentary rocks, a southern segment of Paleozoic and Mesozoic metamorphic rock outcrops and intrusive Mesozoic and Cenozoic batholiths, and a coastal Pacific area of andesitic Mesozoic Jurassic and Cretaceous volcanic-sedimentary rocks [60,61].

## The Basin of Mexico in geological context

The Basin of Mexico, the primary region of interest in this study, is situated in the central-eastern part of the Mexican Volcanic Belt. It is a late Tertiary and Quaternary graben basin characterized by basaltic and andesitic volcanism with single rhyolite cones, featuring some of the most complex volcanic geology of Mexico [61,63,66,67]. The Basin is enclosed by several mountain ranges, including the Sierra de Tepotzotlán and the Sierra de Pachuca to the north, the Sierra de Río Frío and the Sierra Nevada to the east, the Sierra de Chichinautzin to the south, and the Sierra del Ajusco and Sierra de las Cruces to the west.

While the underlying bedrock geology is likely the dominant contribution to the radiogenic strontium isotope composition of the piedmont and mountains of the Basin of Mexico, the alluvial plain represents a large catchment area for weathered minerals deposited by rivers and

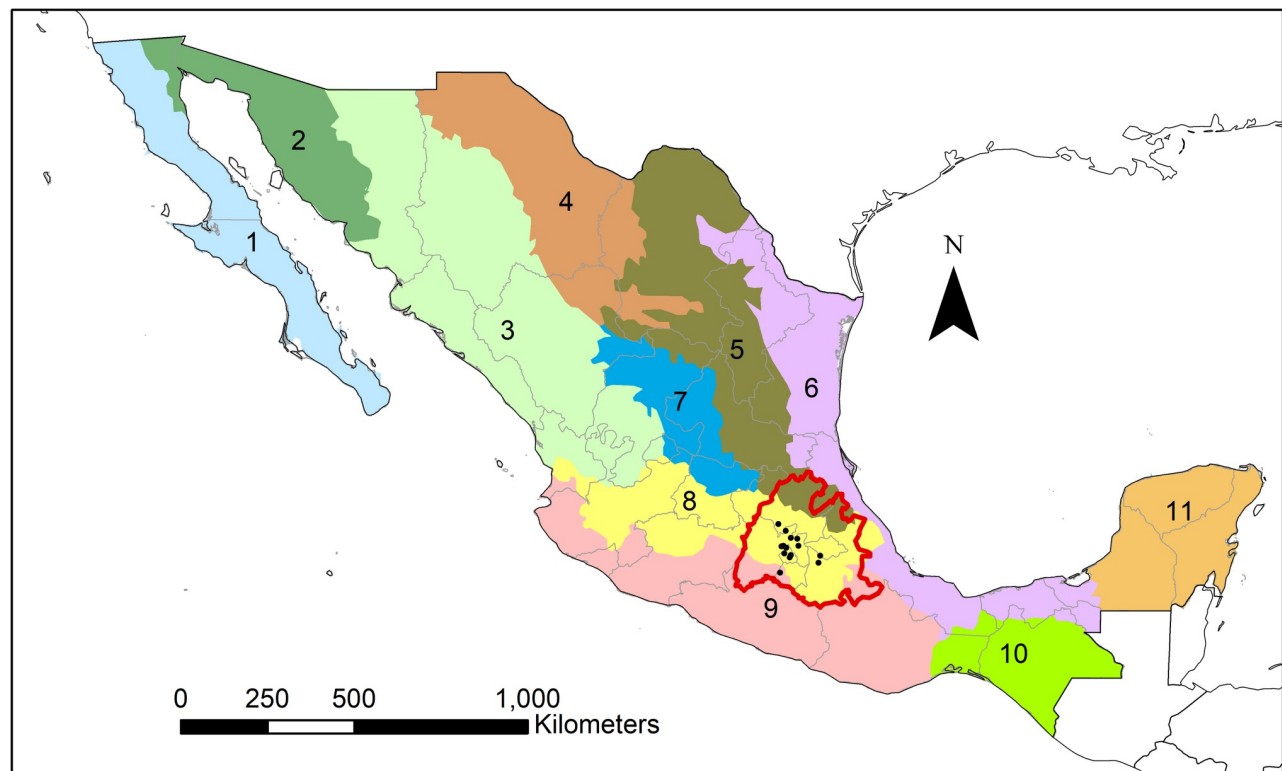

**Fig 1. The morphotectonic provinces of Mexico.** Central Mexico is outlined in red and is made up by parts of the Sierra Madre Oriental (5), the Mexican Volcanic Belt (8), and the Sierra Madre del Sur (9) morphotectonic provinces. Other morphotectonic provinces include Baja California Peninsula (1), the Northwestern Plains and Sierras (2), the Sierra Madre Occidental (3), the Chihuahua-Coahuila Plateaus and Ranges (4), the Gulf Coast Plain (6), the Central Plateau (7), the Sierra Madre de Chiapas (10), and the Yucatán Platform (11). Sites included in the study are indicated by black dots. Map created by SIPF with free vector and raster map data from Natural Earth [57]. Morphotectonic data adapted from the Mexican Geological Service [58].

streams flowing into the Basin lakes. At high elevations, which tend to have high weathering rates, bioavailable $^{87}Sr/^{86}Sr$ and bedrock $^{87}Sr/^{86}Sr$ values are more often closely correlated [68,69]. At lower elevations, however, correlations between underlying bedrock and river content are less clear, as rivers carry suspended loads of upstream rocks and solids as well as precipitation, all of which could contribute geologically distinct strontium values to alluvial deposits [37,40,70]. This suggests that soils in the Basin of Mexico's alluvial plain may vary considerably in strontium isotope values and will likely average source materials. Thus, though the geology of the Basin of Mexico provides starting expectations for ranges of radiogenic strontium variability, it is necessary to generate expected "local" ranges of bioavailable strontium values within the region to gain a more comprehensive understanding of variability within and beyond the Basin of Mexico.

## Materials and methods

### Sample collection

Modern plant and water samples provide an excellent means of characterizing the bioavailable strontium within ecosystems. While soil $^{87}Sr/^{86}Sr$ values in a given geologic zone may vary greatly due to the distinct strontium concentrations and weathering profiles of minerals in the underlying bedrock [34,71], only a proportion of soil strontium is available to plants. As such, plant $^{87}Sr/^{86}Sr$ values provide a consistent average of local bioavailable strontium within a

given ecosystem [72]. Similarly, the majority of strontium in water sources is carried as dissolved or suspended sediment and primarily represents bioavailable strontium from rocks undergoing erosion within an ecosystem [34,37,69,70,73,74].

Plant and water samples were collected between December 2015 and June 2017 from a total of 13 archaeological and agricultural sites from distinct ecological zones throughout the Basin of Mexico and greater central Mexico ($n = 63$). Universal Transverse Mercator (UTM) coordinate and elevation data for each sample were collected using a hand-held GPS unit (S1 File). Plant samples were only collected if it was clear that they had not been treated with fertilizers or irrigation water, as these could skew signatures of local bioavailable strontium with non-local sources of strontium. Furthermore, plants of varied rooting depths were sampled opportunistically. Plants with shallow rooting depths in topsoil (<1 m deep), such as grasses and many herbaceous plants, tend to exhibit $^{87}Sr/^{86}Sr$ values closer to atmospheric dust. In contrast, plants with deeper rooting depths, including many species of tree, exhibit $^{87}Sr/^{86}Sr$ values derived from local bedrock in addition to atmospheric sources [36]. Including both of these sources allows for the more accurate characterization of bioavailable strontium in local ecosystems [75]. Similarly, water samples were only collected from uncontaminated springs that would likely have been used by ancient inhabitants of the region [76,77]. The Mexican Instituto Nacional de Antropología e Historia (INAH) does not require specific permissions to collect water or modern plant samples from the study sites. Furthermore, no endangered or protected plant species were involved in the study. Samples were imported to the Arizona State University Archaeological Chemistry Laboratory under permits granted to Pacheco-Forés from the United States Department of Agriculture Animal and Plant Health Inspection Service (PCIP-17-00469).

Additionally, published central Mexican $^{87}Sr/^{86}Sr$ values generated by Price and colleagues [12] and Schaaf and colleagues [15] were included in the study dataset ($n = 16$). Non-human baseline samples such as soils, plants, or faunal materials [37] were incorporated. Data from published whole rock samples were not included, as these $^{87}Sr/^{86}Sr$ values were likely not bioavailable within the ecosystem. Finally, published data were included only if their provenience could be confirmed via GPS to provide reasonably accurate UTM coordinate and elevation data.

## Biogeochemical methods

All samples were prepared at the Arizona State University Archaeological Chemistry Laboratory. Water samples were filtered (2.5μm diameter) and acidified to 5% HCl to prevent precipitates from forming, adsorbtion to bottle walls, and discourage bacterial and algal growth. When possible, pre-Hispanic diets were simulated through the manual isolation and analysis of edible components (e.g., seeds, berries, leaves) of dried plants [78]. Plant samples were rinsed with 18.2 MΩ Millipore water to remove adhering dirt and were ashed in a furnace for approximately 10 hours at 800° C. Approximately 25.0 mg of ashed sample was digested in 2 mL of concentrated nitric and hydrochloric acid ($HNO_3 + 3HCl$) at approximately 50° C for 24 hours. This aggressive leach does not break down the silica tetrahedra structure of most silicate minerals, leaving much of the soil in a solid form while prioritizing the release of bioavailable strontium within plants. Leach solution was evaporated, and sample precipitates were redissolved in concentrated nitric acid and diluted to a 2 M stock solution.

Dissolved samples were analyzed at the Metals, Environmental, and Terrestrial Analytical Laboratory at Arizona State University. An aliquot was taken for elemental concentration by a Thermo Fisher Scientific iCAP quadrupole inductively coupled plasma mass spectrometer (Q-ICP-MS). Strontium was then separated with a Prep*FAST*, an automated low-pressure ion

exchange chromatography system [79]. Strontium was isolated from the sample matrix using Elemental Scientific, Inc. supplied Sr-Ca ion exchange resin (Part CF-MC-SrCa-1000) and ultrapure 5 M nitric acid ($HNO_3$). Each strontium cut from the Prep*FAST* was dried down in a Teflon beaker and digested with concentration nitric acid and 30% hydrogen peroxide to remove organics from the resin. Once digested, samples were again dried down and reconstituted with 0.32 M nitric acid. Using concentration information from the Q-ICP-MS, the samples were diluted with 0.32 M nitric acid to a calculated constant concentration of 50 ppb Sr.

Radiogenic strontium isotope ratios were measured on a Thermo-Finnigan Neptune multi-collector inductively coupled plasma mass spectrometer (MC-ICP-MS). The MC-ICP-MS has nine Faraday cups capable of simultaneous ion beam measurement, and this instrument was configured with an Elemental Scientific, Inc., Apex Q high sensitivity sample introduction system with an Elemental Scientific, Inc. 50 or 100 μL/minute PFA-ST microflow nebulizer. This instrument has seven 1011 amplifiers and three 1012 amplifiers which can be designated for any of the Faraday cups.

Data was collected by measuring 60 simultaneous ratios integrating 4.194 seconds each. Samples were corrected for on-peak blanks, and in-line correction of the contributions of $^{84}$Kr on $^{84}$Sr and $^{86}$Kr on $^{86}$Sr using $^{83}$Kr/$^{84}$Kr ratio of 0.201750 and $^{83}$Kr/$^{86}$Kr ratio of 0.664533, after instrumental mass bias correction using a normalizing $^{88}$Sr/$^{86}$Sr ratio of 8.375209. Samples were analyzed in three different analytical sessions. Typical sensitivity was >10 V on $^{88}$Sr with a 50 ppb Sr solution, with $^{83}$Kr values <0.0001 V. $^{85}$Rb voltages for samples were typically <0.004 V due to the low Rb/Sr initial ratios of the samples and effective chemical purification, but all data was interference-corrected using a $^{85}$Rb/$^{87}$Rb ratio of 2.588960, normalized to $^{88}$Sr/$^{86}$Sr as above. Ratio outliers two standard deviations outside the mean were removed using a Matlab 2D-mathematical correction routine written by Dr. Stephen Romaniello, now at University of Tennessee. Typical internal $^{87}$Sr/$^{86}$Sr two standard error (SE) precision was ~1e-6.

Sequences included bracketing concentration-matched SRM 987 standards. SRM 987 was run as a bracketing standard with a measured value of $^{87}$Sr/$^{86}$Sr = 0.710252 ±0.000026 (2σ, $n$ = 89). Each analytical session included a sequence incorporating SRM 987 standard in a range of variable concentrations to verify the accuracy of $^{87}$Sr/$^{86}$Sr values for samples; reported values are all above the threshold for accurate $^{87}$Sr/$^{86}$Sr values within the range of error of the bracketing standards. In addition, SRM 987 doped with calcium up to a ratio of Ca/Sr of 500 was run to simulate the accuracy and precision of isotope ratios in poorly purified samples with low yields. SRM 987 run at 50% concentration doped to a Ca/Sr of 500 was run as a check standard with a measured value of $^{87}$Sr/$^{86}$Sr = 0.710253 ± 0.000025 (2σ, $n$ = 15). IAPSO seawater (Ocean Scientific International Ltd., Havant, UK) as a secondary check standard had a measured value of 0.709182 ±0.000010 (2σ, $n$ = 11), within error of the published value of 0.709182 ±0.000004 [80]. NIST 1400 purified in parallel with samples had a measured value of 0.713124 ±0.000023 (2σ, $n$ = 12), similar to the published value of 0.713150 ±0.0000160 [81].

## Analytical methods

K-means cluster analysis was used to sort observed and published $^{87}$Sr/$^{86}$Sr, UTM, and elevation data into groups in R using the cluster and ggplot2 packages [82–84]. K-means cluster analysis is a divisive iterative non-hierarchical pure locational clustering method [85,86] that has been applied to the analysis of analysis of bioavailable $^{87}$Sr/$^{86}$Sr isotopes [40]. Clusters were defined based on Euclidean distances to minimize the sum of squares error (SSE), thus minimizing variability within clusters while maximizing variability between clusters. A randomization procedure assessing changes in the global SSE for different cluster levels was conducted. A

cluster solution was selected by comparing the difference in SSE in the original data to the mean SSE of 1,000 randomized iterations of the data (S1 and S2 Files).

## Results and discussion

Table 1 reports observed and published $^{87}$Sr/$^{86}$Sr values of water, plant, faunal, and soil samples included the study. $^{87}$Sr/$^{86}$Sr values varied from 0.70432 to 0.70641. Among plant samples, opportunistically sampled non-native and non-edible plants did not provide significantly different values from native edible plants simulating pre-Hispanic diets (S1 Fig). All generated trace elemental concentration data from the Q-ICP-MS (S1 Appendix) and radiogenic strontium data from the MC-ICP-MS (S2 Appendix) are available as supplementary spreadsheets.

The randomization procedure indicates a five-cluster solution represents the greatest departure in the global SSE from randomness. The data are not normally distributed. Medians and interquartile ranges are therefore used to characterize $^{87}$Sr/$^{86}$Sr variability within each cluster, following Price and colleagues [18] (Table 2, Figs 2 and 3). In cases where sites (S1 Table) or clusters have fewer than three samples, simple ranges are provided in lieu of interquartile ranges.

Each of the five clusters form culturally meaningful geographically distinct subregions within central Mexico (Fig 2). Cluster 1 is made up of two sites north of the Basin of Mexico. The Basin of Mexico itself is divided into two clusters, Cluster 2 which comprises the northeast of the Basin (three sites), and Cluster 3 which makes up the southwest of the Basin (seven sites). Cluster 4 is comprised of two sites in the Puebla-Tlaxcala Valley, and Cluster 5 is made up of the site of Xochicalco, south of the Basin of Mexico. Overall, cluster $^{87}$Sr/$^{86}$Sr ranges conform to geologic expectations. The Basin of Mexico clusters have the lowest $^{87}$Sr/$^{86}$Sr values, reflecting the Basin's origins in Cenozoic volcanism [67,87]. In contrast, the Xochicalco cluster has the highest $^{87}$Sr/$^{86}$Sr values, indicating the region's Mesozoic origins [87,88], although the intra-region variability is poorly constrained given the number of data points ($n = 4$). Finally, the Puebla-Tlaxcala Valley cluster has intermediate values consistent with the region's Mesozoic platforms overlain by Cenozoic volcanic rocks [89].

While the five-cluster model divides the Basin into two distinct groups, it is notable that there is significant overlap in $^{87}$Sr/$^{86}$Sr values between Basin clusters, as well as with $^{87}$Sr/$^{86}$Sr values in the cluster north of the Basin (Fig 3). Interestingly, $^{87}$Sr/$^{86}$Sr values of the southwest Basin of Mexico cluster are most variable within the Basin of Mexico. This may reflect the greater diversity in age of the geologic substrate, as the southwestern Basin is made up by some of the oldest and youngest geologic formations in the Basin, including the Xochitepec Formation (Oligocene, 33.9–23.0 Ma) and the Chichinautzin mountain range (Quaternary, 2.6 Ma-present). Despite overlapping ranges among Basin of Mexico clusters, $^{87}$Sr/$^{86}$Sr interquartile ranges indicate that sites in the Basin of Mexico are readily distinguishable from those in the Puebla-Tlaxcala Valley to the east, as well as Xochicalco to the south. Radiogenic strontium isotopes can thus be used to address questions of paleomobility at the regional level within central Mexico.

The generated Basin of Mexico interquartile range is consistent with previously published ranges. The two Basin of Mexico clusters (2–3) have a combined interquartile range of $^{87}$Sr/$^{86}$Sr = 0.70465–0.70487 ($n = 60$). While this range is consistent with the $^{87}$Sr/$^{86}$Sr = 0.7046–0.7051 ($n = 86$) published by Price and colleagues [18], examination of site-specific $^{87}$Sr/$^{86}$Sr interquartile ranges indicates that this local range belies a great deal of variability within the Basin. Many sites in Basin of Mexico clusters can still be distinguished using radiogenic strontium analysis (Fig 4, S1 Table). Furthermore, with a few notable exceptions, including Teotihuacan in the northeast Basin cluster and Cuicuilco and Tezozomoc in the southwest

**Table 1.** $^{87}Sr/^{86}Sr$ and provenance data from Basin of Mexico and greater central Mexico baseline samples.

| Laboratory Number | Site | Material | $^{87}Sr/^{86}Sr$ | UTM-E | UTM-N | Altitude (masl) | Cluster |
|---|---|---|---|---|---|---|---|
| ACL-7409-FT | Tequixquiac, Mexico State | spring water | 0.70476 | 484002 | 2199133 | 2239 | 1 |
| ACL-7409-UF | Tequixquiac, Mexico State | spring water | 0.70469 | 484002 | 2199133 | 2239 | 1 |
| ACL-7410-FT | Tequixquiac, Mexico State | spring water | 0.70462 | 480117 | 2200273 | 2533 | 1 |
| ACL-7410-UF | Tequixquiac, Mexico State | spring water | 0.70458 | 480117 | 2200273 | 2533 | 1 |
| TU-1S | Tula, Hidalgo | soil[a] | 0.70500 | 464348 | 2218555 | 2050 | 1 |
| TU-2S | Tula, Hidalgo | soil[a] | 0.70501 | 464348 | 2218555 | 2050 | 1 |
| TU-3S | Tula, Hidalgo | soil[a] | 0.70469 | 464348 | 2218555 | 2050 | 1 |
| ACL-9058 | Texcotzingo, Mexico State | Opuntia ficus | 0.70471 | 519433 | 2155797 | 2513 | 2 |
| ACL-9059 | Texcotzingo, Mexico State | Dahlia pinnata | 0.70459 | 519358 | 2155730 | 2504 | 2 |
| ACL-9060 | Texcotzingo, Mexico State | Agave spp. | 0.70464 | 519020 | 2155859 | 2534 | 2 |
| ACL-7374 | Xaltocan, Mexico State | Kochia scoparia | 0.70480 | 495867 | 2178713 | 2239 | 2 |
| ACL-7375 | Xaltocan, Mexico State | Poa spp. | 0.70479 | 495867 | 2178716 | 2239 | 2 |
| ACL-7376 | Xaltocan, Mexico State | Poa spp. | 0.70480 | 495878 | 2178710 | 2239 | 2 |
| ACL-7377 | Xaltocan, Mexico State | Chenopodium nuttalliae | 0.70482 | 495882 | 2178707 | 2239 | 2 |
| ACL-7378 | Xaltocan, Mexico State | Chenopodium nuttalliae | 0.70480 | 495883 | 2178708 | 2239 | 2 |
| ACL-7379 | Xaltocan, Mexico State | Chenopodium nuttalliae | 0.70479 | 495892 | 2178710 | 2239 | 2 |
| ACL-7380 | Xaltocan, Mexico State | Avena sativa | 0.70477 | 495702 | 2178935 | 2238 | 2 |
| ACL-7381 | Xaltocan, Mexico State | Helianthus spp. | 0.70479 | 494737 | 2178926 | 2238 | 2 |
| ACL-7382 | Xaltocan, Mexico State | Avena sativa | 0.70478 | 494837 | 2178943 | 2239 | 2 |
| ACL-7383 | Xaltocan, Mexico State | Taraxacum officinale | 0.70474 | 497854 | 2178943 | 2239 | 2 |
| ACL-7384 | Xaltocan, Mexico State | Taraxacum officinale | 0.70475 | 495063 | 2178959 | 2239 | 2 |
| ACL-7385 | Xaltocan, Mexico State | Hordeum vulgare | 0.70478 | 495346 | 2178978 | 2239 | 2 |
| ACL-7386 | Xaltocan, Mexico State | Chenopodium nuttalliae | 0.70481 | 495347 | 2178979 | 2239 | 2 |
| ACL-7387 | Xaltocan, Mexico State | Hordeum vulgare | 0.70484 | 495423 | 2178904 | 2239 | 2 |
| ACL-7388 | Xaltocan, Mexico State | Poa spp. | 0.70484 | 495637 | 2178998 | 2239 | 2 |
| ACL-7389 | Xaltocan, Mexico State | Jaltomata procumbens | 0.70497 | 495884 | 2178860 | 2239 | 2 |
| ACL-7390 | Xaltocan, Mexico State | Poa spp. | 0.70488 | 495868 | 2178714 | 2239 | 2 |
| ACL-7391 | Xaltocan, Mexico State | Poa spp. | 0.70490 | 495868 | 2178713 | 2239 | 2 |
| ACL-7394 | Xaltocan, Mexico State | Helianthus spp. | 0.70481 | 495846 | 2178692 | 2238 | 2 |
| ACL-7397 | Xaltocan, Mexico State | Kochia scoparia | 0.70482 | 495826 | 2178687 | 2238 | 2 |
| ACL-7399 | Xaltocan, Mexico State | Agave spp. | 0.70471 | 495251 | 2181200 | 2241 | 2 |
| ACL-7400 | Xaltocan, Mexico State | Opuntia ficus | 0.70490 | 495292 | 2181209 | 2242 | 2 |
| 11203 CV C2 N334 E96 11 | Teotihuacan, Mexico State | Sylvilagus spp.[b] | 0.70459 | 516371 | 2177462 | 2351 | 2 |
| 11145 CV C2 N331 E93 1k | Teotihuacan, Mexico State | Sylvilagus spp.[b] | 0.70458 | 516371 | 2177462 | 2351 | 2 |
| 3110 CV C1 N342 E94 1a | Teotihuacan, Mexico State | Sylvilagus spp.[b] | 0.70468 | 516371 | 2177462 | 2351 | 2 |
| 8186 CV T N333 E81 2d | Teotihuacan, Mexico State | Sylvilagus spp.[b] | 0.70459 | 516371 | 2177462 | 2351 | 2 |
| 3294 CV C1 N338 E91 1a | Teotihuacan, Mexico State | Sylvilagus spp.[b] | 0.70464 | 516371 | 2177462 | 2351 | 2 |
| 7531 CV NS N334 E91 1a | Teotihuacan, Mexico State | Sylvilagus spp.[b] | 0.70471 | 516371 | 2177462 | 2351 | 2 |
| 22422 CP C5 N348 E116 1f/2a | Teotihuacan, Mexico State | Sylvilagus spp.[b] | 0.70461 | 516371 | 2177462 | 2351 | 2 |
| 790 CB N325 E16 S | Teotihuacan, Mexico State | Sylvilagus spp.[b] | 0.70470 | 516371 | 2177462 | 2351 | 2 |
| 706 CB N332 E31 S | Teotihuacan, Mexico State | Sylvilagus spp.[b] | 0.70465 | 516371 | 2177462 | 2351 | 2 |
| 67145s | Teotihuacan, Mexico State | soil[a] | 0.70435 | 516371 | 2177462 | 2351 | 2 |
| 67145s | Teotihuacan, Mexico State | soil[a] | 0.70432 | 516371 | 2177462 | 2351 | 2 |
| 25166s | Teotihuacan, Mexico State | soil[a] | 0.70438 | 516371 | 2177462 | 2351 | 2 |
| 25166s | Teotihuacan, Mexico State | soil[a] | 0.70441 | 516371 | 2177462 | 2351 | 2 |
| ACL-9046 | Cuicuilco, Mexico City | Agave spp. | 0.70507 | 480790 | 2134234 | 2290 | 3 |
| ACL-9047 | Cuicuilco, Mexico City | Dahlia pinnata | 0.70536 | 480998 | 2134251 | 2288 | 3 |

*(Continued)*

**Table 1.** (Continued)

| Laboratory Number | Site | Material | $^{87}$Sr/$^{86}$Sr | UTM-E | UTM-N | Altitude (masl) | Cluster |
|---|---|---|---|---|---|---|---|
| ACL-9048 | Cuicuilco, Mexico City | *Verbascum giganteum* | 0.70502 | 481044 | 2134137 | 2283 | 3 |
| ACL-9049 | Cuicuilco, Mexico City | *Opuntia ficus* | 0.70591 | 480991 | 2134066 | 2286 | 3 |
| ACL-9050 | Tezozomoc, Mexico City | *Schinus molle* | 0.70520 | 477880 | 2156155 | 2251 | 3 |
| ACL-9051 | Tezozomoc, Mexico City | *Agave* spp. | 0.70618 | 478014 | 2156268 | 2251 | 3 |
| ACL-9052 | Naucalli, Mexico State | *Yucca filifera* | 0.70497 | 474873 | 2155369 | 2264 | 3 |
| ACL-9053 | Naucalli, Mexico State | *Opuntia ficus* | 0.70503 | 475008 | 2155777 | 2264 | 3 |
| ACL-9054 | Naucalli, Mexico State | *Agave* spp. | 0.70469 | 474863 | 2155595 | 2264 | 3 |
| ACL-9055 | Cerro Moctezuma, Mexico State | *Arctostaphylos* spp. | 0.70455 | 473040 | 2154358 | 2385 | 3 |
| ACL-9056 | Cerro Moctezuma, Mexico State | *Agave* spp. | 0.70489 | 472950 | 2154408 | 2397 | 3 |
| ACL-9057 | Cerro Moctezuma, Mexico State | *Dahlia pinnata* | 0.70471 | 473033 | 2154438 | 2382 | 3 |
| ACL-9061 | Tlatelolco, Mexico City | *Poa* spp. | 0.70484 | 485501 | 2150723 | 2231 | 3 |
| ACL-9062 | Tlatelolco, Mexico City | *Yucca filifera* | 0.70483 | 485523 | 2150718 | 2231 | 3 |
| ACL-9063 | Tlatelolco, Mexico City | *Opuntia ficus* | 0.70496 | 485597 | 2150719 | 2233 | 3 |
| ACL-9064 | Tlatelolco, Mexico City | *Agave* spp. | 0.70514 | 485543 | 2150786 | 2233 | 3 |
| ACL-9069 | San Pedro Atocpan, Mexico City | *Opuntia ficus* | 0.70486 | 494693 | 2122995 | 2239 | 3 |
| ACL-9070 | San Pedro Atocpan, Mexico City | *Poa* spp. | 0.70481 | 494693 | 2122995 | 2239 | 3 |
| ACL-9071 | San Pedro Atocpan, Mexico City | *Amaranthus hybridus* | 0.70452 | 494693 | 2122995 | 2239 | 3 |
| ACL-9072 | Santiago Tulyehualco, Mexico City | *Pinus* spp. | 0.70466 | 498319 | 2128955 | 2252 | 3 |
| ACL-9073 | Santiago Tulyehualco, Mexico City | *Agave* spp. | 0.70455 | 498319 | 2128955 | 2252 | 3 |
| ACL-9074 | Santiago Tulyehualco, Mexico City | *Agave* spp. | 0.70462 | 498319 | 2128955 | 2252 | 3 |
| ACL-9075 | Cholula, Puebla | *Dorotheanthus* spp. | 0.70543 | 573378 | 2107043 | 2148 | 4 |
| ACL-9076 | Cholula, Puebla | *Opuntia ficus* | 0.70598 | 573421 | 2107130 | 2150 | 4 |
| ACL-9077 | Cholula, Puebla | *Chenopodium nuttalliae* | 0.70575 | 573314 | 2107191 | 2154 | 4 |
| ACL-9078 | Cholula, Puebla | *Agave* spp. | 0.70602 | 573258 | 2107461 | 2157 | 4 |
| ACL-9079 | Cacaxtla, Tlaxcala | *Dahlia pinnata* | 0.70500 | 569529 | 2127993 | 2298 | 4 |
| ACL-9080 | Cacaxtla, Tlaxcala | *Agave* spp. | 0.70553 | 569461 | 2128024 | 2302 | 4 |
| ACL-9081 | Cacaxtla, Tlaxcala | *Quercus* spp. | 0.70525 | 569280 | 2128072 | 2305 | 4 |
| ACL-9082 | Cacaxtla, Tlaxcala | *Agave* spp. | 0.70541 | 569395 | 2127878 | 2309 | 4 |
| ACL-9065 | Xochicalco, Morelos | *Agave* spp. | 0.70641 | 468747 | 2079174 | 1349 | 5 |
| ACL-9066 | Xochicalco, Morelos | *Enterolobium cyclocarpum* | 0.70521 | 468921 | 2079148 | 1329 | 5 |
| ACL-9067 | Xochicalco, Morelos | *Agave* spp. | 0.70600 | 468629 | 2079292 | 1348 | 5 |
| ACL-9068 | Xochicalco, Morelos | *Agave* spp. | 0.70539 | 468872 | 2079236 | 1340 | 5 |

[a] Data from bulk soil samples published in [15]

[b] Data published in [12]

Basin cluster, all site-specific $^{87}$Sr/$^{86}$Sr "local" ranges are narrower than the $^{87}$Sr/$^{86}$Sr ranges of their assigned clusters. This suggests that while the k-means cluster analysis is useful on a

**Table 2.** $^{87}$Sr/$^{86}$Sr medians and interquartile ranges for central Mexican subregions identified through k-means cluster analysis.

| Cluster | Geographic Subregion | Median $^{87}$Sr/$^{86}$Sr | Interquartile $^{87}$Sr/$^{86}$Sr Range | | *n* |
|---|---|---|---|---|---|
| 1 | North of the Basin of Mexico | 0.70469 | 0.70466 - | 0.70488 | 7 |
| 2 | Basin of Mexico Northeast | 0.70476 | 0.70464 - | 0.70481 | 38 |
| 3 | Basin of Mexico Southwest | 0.70488 | 0.70470 - | 0.70506 | 22 |
| 4 | Puebla-Tlaxcala Valley | 0.70548 | 0.70537 - | 0.70581 | 8 |
| 5 | Xochicalco | 0.70570 | 0.70535 - | 0.70610 | 4 |

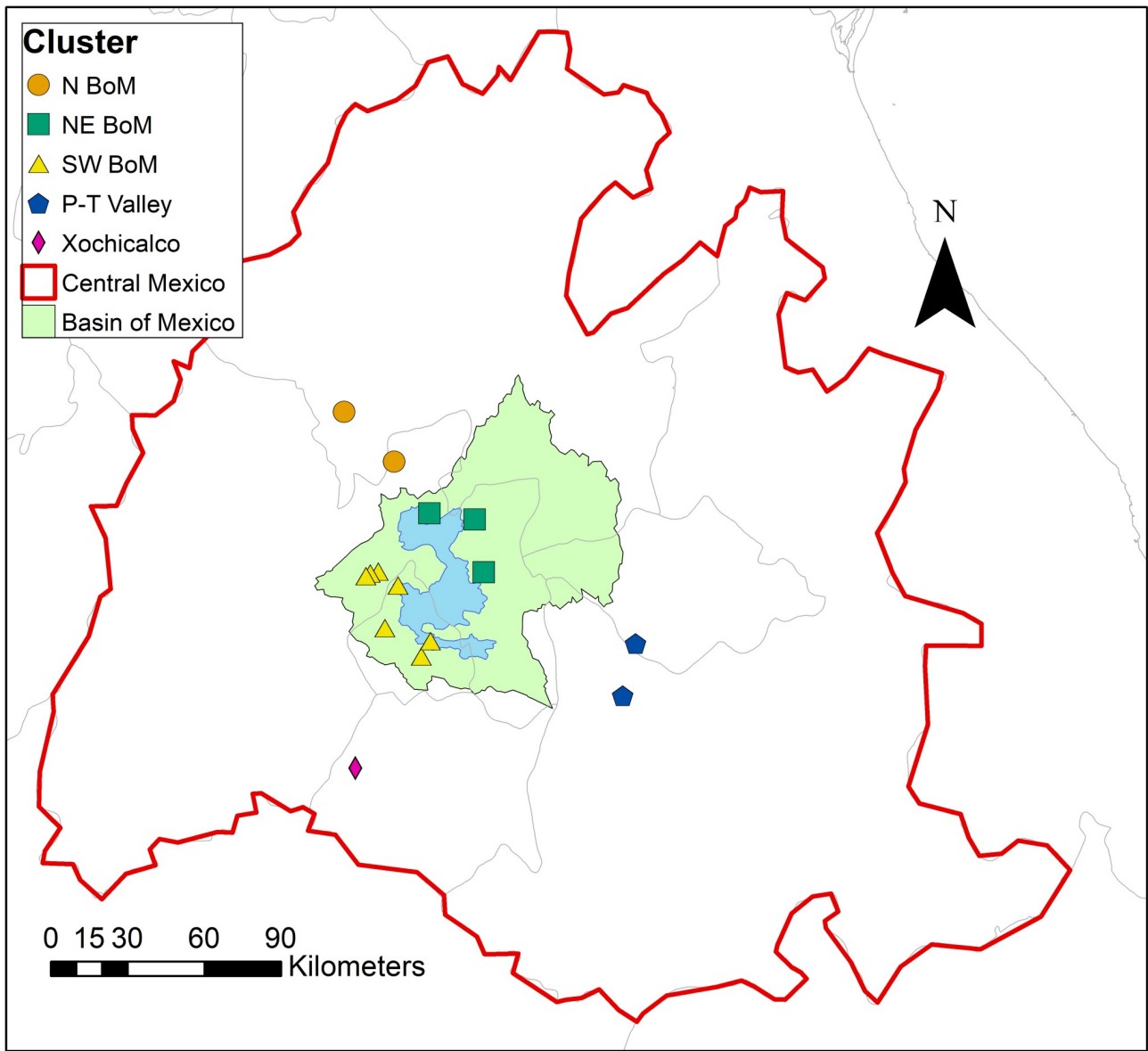

**Fig 2. Sampled sites within central Mexico sorted by cluster membership.** The Basin of Mexico is highlighted in green, and the extinct highland lake system is shown in blue. Map created by SIPF with free vector and raster map data from Natural Earth [57].

larger scale for isotopically distinguishing the Basin of Mexico from surrounding regions within central Mexico, it does not perform well dividing the Basin itself into isotopically distinct subregions.

In the context of paleomobility studies, the use of cluster (Table 2) or site-specific (S1 Table) $^{87}Sr/^{86}Sr$ interquartile ranges as a "local" bioavailable baseline should be determined by the scale of the research question. For example, if a study seeks to identify individuals who migrated into the Basin of Mexico from greater central Mexico and beyond, using cluster "local" $^{87}Sr/^{86}Sr$ ranges provides a robust mechanism for establishing individuals as non-locals within the Basin of Mexico. If, however, a study seeks to identify an individual's residential mobility within the Basin of Mexico, using site-specific "local" $^{87}Sr/^{86}Sr$ ranges will provide a higher resolution analysis. With all such analyses, it is important to keep in mind that $^{87}Sr/^{86}Sr$

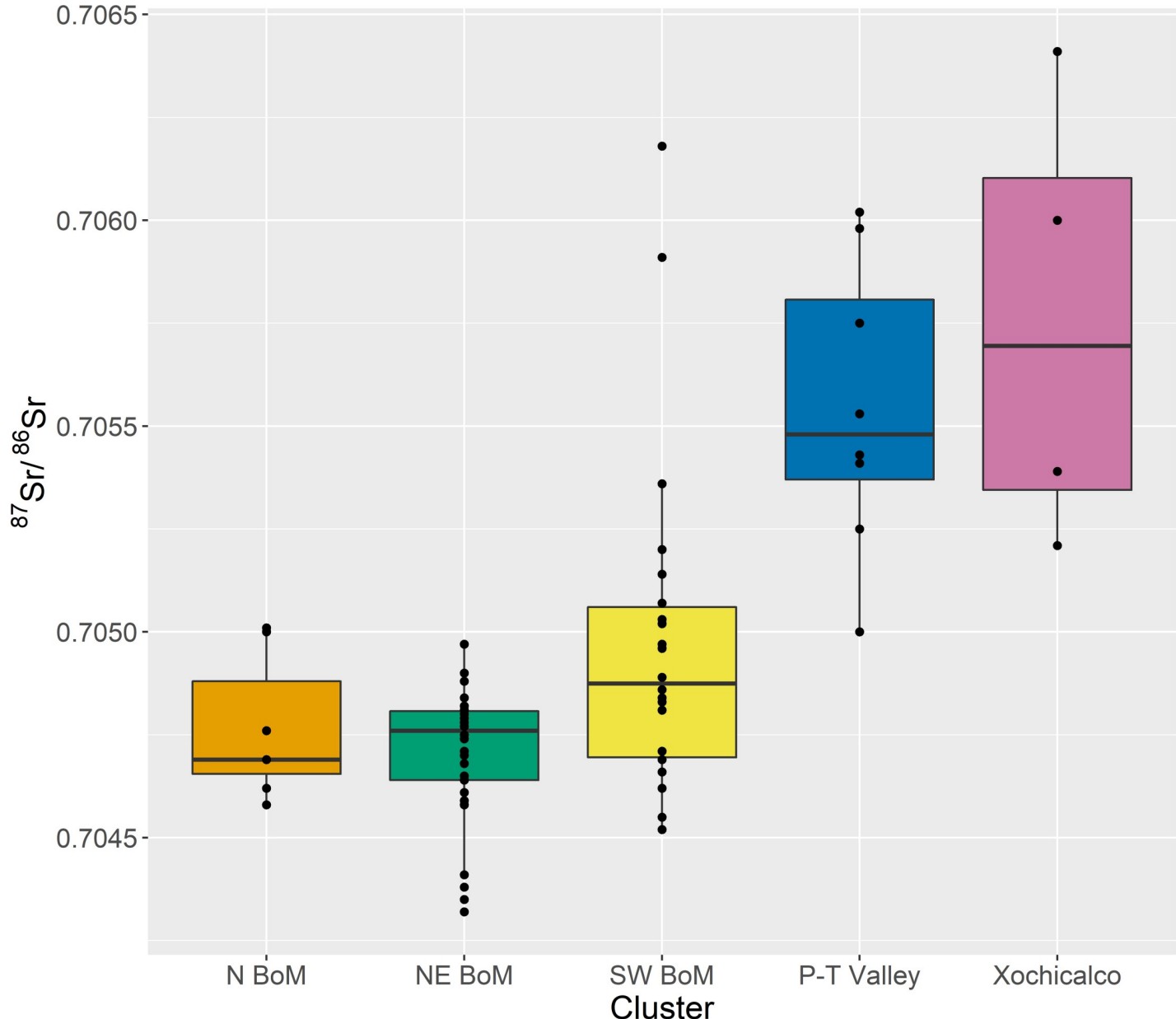

**Fig 3. Medians and interquartile ranges (filled) of each clustered subregion in Table 2, with superimposed individual data points.** BoM = Basin of Mexico, P-T = Puebla-Tlaxcala.

values are not unique and may mask the presence of non-locals if these individuals were from a region with similar $^{87}Sr/^{86}Sr$ values. For this reason, the use of multiple lines of evidence and isotopic systems is essential [13,51,90].

## Conclusion

Analysis of presented and published bioavailable radiogenic strontium isotope ratios from central Mexico indicates that the Basin of Mexico can be distinguished isotopically from neighboring central Mexican regions. Furthermore, many sites within the Basin of Mexico itself can be distinguished from each other using radiogenic strontium isotopes, despite some overlap in

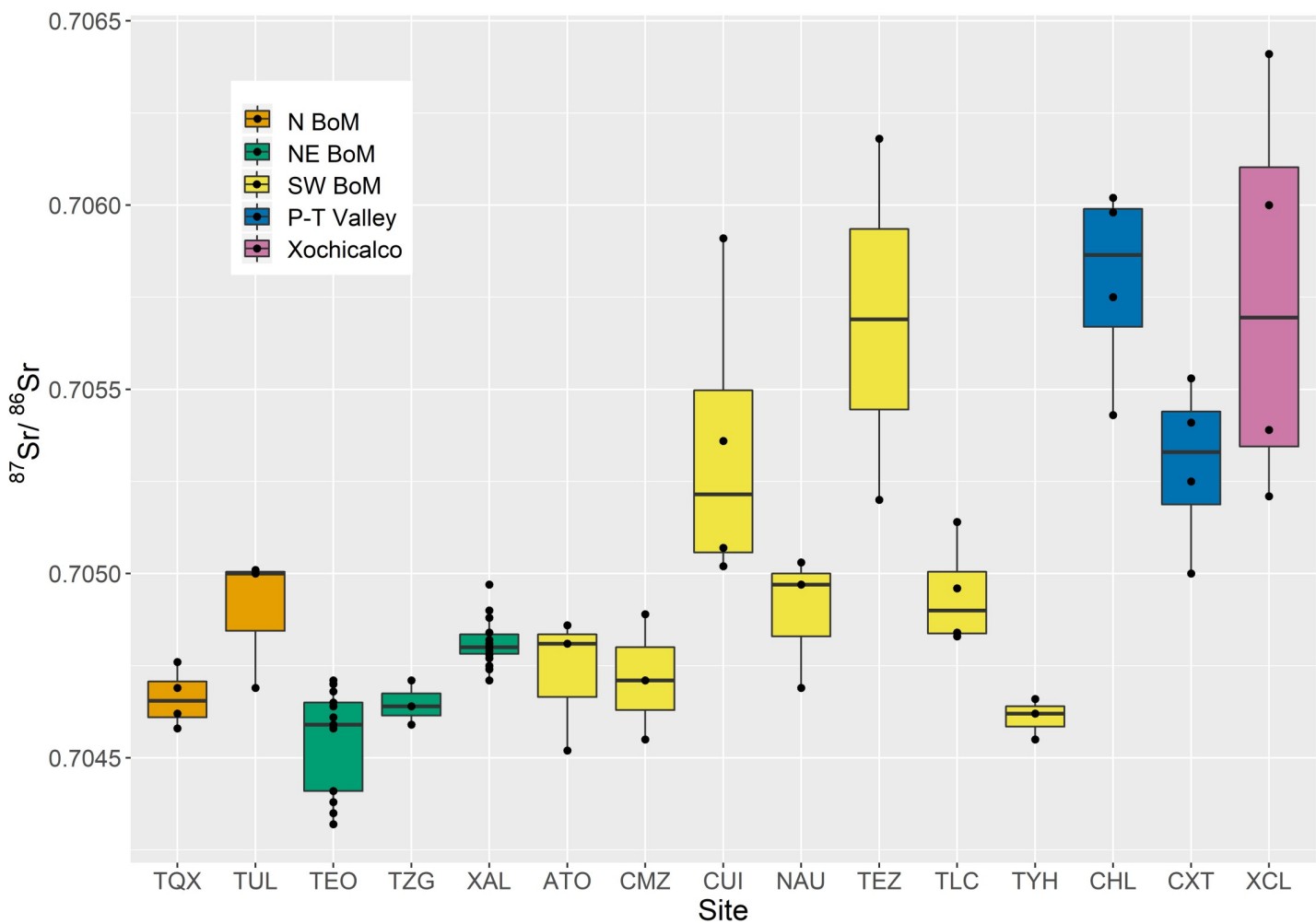

**Fig 4. $^{87}$Sr/$^{86}$Sr interquartile ranges of central Mexican sites, shaded by cluster.** Individual data points are overlain. TQX = Tequixquiac, TUL = Tula, TEO = Teotihuacan, TZG = Texcotzingo, XAL = Xaltocan, ATO = San Pedro Atocpan, CMZ = Cerro Moctezuma, CUI = Cuicuilco, NAU = Naucalli, TEZ = Tezozomoc, TLC = Tlatelolco, THY = Santiago Tulyehualco, CHL = Cholula, CXT = Cacaxtla, XCL = Xochicalco.

$^{87}$Sr/$^{86}$Sr cluster expected local ranges. This indicates that radiogenic strontium isotopes remain a powerful tool for examining paleomobility within central Mexico, particularly if used in concert with other isotopic systems, such as oxygen ($\delta^{18}$O) [91].

Expanding knowledge of radiogenic strontium isotope variability within central Mexico is essential for future paleomobility work in the region, particularly given the hypothesized importance of migration in the cultural development of the region [3,92]. Future work will focus on augmenting the baseline data presented here with samples from additional sites throughout greater central Mexico. These data will be stored in an open-access comprehensive database of strontium isotopes throughout central Mexico with the ultimate goal of developing an $^{87}$Sr/$^{86}$Sr isoscape for the region.

## Supporting information

**S1 File. CSV data spreadsheet to load into R for use with code in S2 File.**
(CSV)

**S2 File. R code for statistical analysis of $^{87}$Sr/$^{86}$Sr, UTM coordinate, and elevation data.**
(RMD)

**S1 Fig. Cluster $^{87}$Sr/$^{86}$Sr values in plant samples by plant origin.** There were no significant differences between edible native plants and non-edible native plants or non-native plants. While non-edible native and non-native plants would not have contributed to past human and animal bioavailable $^{87}$Sr/$^{86}$Sr values, they are included in this study to further characterize bio-available strontium values in local ecosystems.
(TIF)

**S1 Appendix. Generated trace elemental concentration data from the Q-ICP-MS in central Mexican plant and water samples.**
(XLSX)

**S2 Appendix. Generated $^{87}$Sr/$^{86}$Sr values from the MC-ICP-MS in central Mexican plant and water samples.**
(XLSX)

**S1 Table. Central Mexican site-level $^{87}$Sr/$^{86}$Sr medians and interquartile ranges.**
(DOCX)

**S1 Translation. Spanish language translation of the present manuscript.**
(DOCX)

## Acknowledgments

We are thankful to Andrés Mejía-Ramón, Dr. Christopher Morehart, Camila Pacheco-Forés, and Edgar Paredes for assistance with sample collection in Mexico. At the Archaeological Chemistry Laboratory, we are thankful to research apprentices Aimee Alvarado, Jorge Benavente, Sibella Campbell, Eric Flores, Zen Garcia, Kari Guilbault, Arman Gurule, Sparshee Naik, Elizabeth Rausch, Emily Steinberg, Alyssa Torres, Rebecca Ulloa, and Tajinder Virdee. At the W. M. Keck Foundation Laboratory for Environmental Biogeochemistry, we are grateful for the assistance of Dr. Stephen Romaniello, Dr. Trevor Martin, and Natasha Zolotova. We thank Dr. Christina Stantis and one anonymous reviewer for providing insightful comments that improved the clarity of the manuscript.

## Author Contributions

**Conceptualization:** Sofía I. Pacheco-Forés.

**Data curation:** Sofía I. Pacheco-Forés, Gwyneth W. Gordon, Kelly J. Knudson.

**Formal analysis:** Sofía I. Pacheco-Forés, Gwyneth W. Gordon.

**Funding acquisition:** Sofía I. Pacheco-Forés.

**Investigation:** Sofía I. Pacheco-Forés.

**Methodology:** Gwyneth W. Gordon, Kelly J. Knudson.

**Project administration:** Sofía I. Pacheco-Forés.

**Resources:** Sofía I. Pacheco-Forés, Gwyneth W. Gordon, Kelly J. Knudson.

**Supervision:** Kelly J. Knudson.

**Visualization:** Sofía I. Pacheco-Forés.

**Writing – original draft:** Sofía I. Pacheco-Forés.

**Writing – review & editing:** Gwyneth W. Gordon, Kelly J. Knudson.

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
