## [Decision Letter · Decision Letter 0]

31 Jan 2020

PONE-D-19-30394

Expanding radiogenic strontium isotope baseline data for central Mexican paleomobility studies

PLOS ONE

Dear Ms. Pacheco-Fores,

Thank you for submitting your manuscript to PLOS ONE. After careful consideration, we feel that it has merit but does not fully meet PLOS ONE’s publication criteria as it currently stands. Therefore, we invite you to submit a revised version of the manuscript that addresses the points raised during the review process.

Both reviewers have provided comments on the presentation and analyses of your results that I hope you find useful in your revision. In particular they would like clarification on sampling used, and models used in analyses of the data.

Please refer to each of the reviewer's points in your revision. 

We would appreciate receiving your revised manuscript by Mar 16 2020 11:59PM. To enhance the reproducibility of your results, we recommend that if applicable you deposit your laboratory protocols in protocols.io, where a protocol can be assigned its own identifier (DOI) such that it can be cited independently in the future. For instructions see: http://journals.plos.org/plosone/s/submission-guidelines#loc-laboratory-protocols

We look forward to receiving your revised manuscript.

Kind regards,

Siân E Halcrow, Ph.D.

Academic Editor

PLOS ONE

Journal Requirements:

2. In your Methods section, please provide additional location information, including geographic coordinates for the data set if available.

3. We note that Figures 1 and 2 in your submission contain map images which may be copyrighted. All PLOS content is published under the Creative Commons Attribution License (CC BY 4.0), which means that the manuscript, images, and Supporting Information files will be freely available online, and any third party is permitted to access, download, copy, distribute, and use these materials in any way, even commercially, with proper attribution. For these reasons, we cannot publish previously copyrighted maps or satellite images created using proprietary data, such as Google software (Google Maps, Street View, and Earth). For more information, see our copyright guidelines: http://journals.plos.org/plosone/s/licenses-and-copyright.

1.    You may seek permission from the original copyright holder of Figures 1 and 2 to publish the content specifically under the CC BY 4.0 license. 

Reviewers' comments:

Reviewer's Responses to Questions

**Comments to the Author**

1. Is the manuscript technically sound, and do the data support the conclusions?

Reviewer #1: Yes

Reviewer #2: Yes

2. Has the statistical analysis been performed appropriately and rigorously? 

Reviewer #1: Yes

Reviewer #2: Yes

3. Have the authors made all data underlying the findings in their manuscript fully available?

Reviewer #1: Yes

Reviewer #2: Yes

4. Is the manuscript presented in an intelligible fashion and written in standard English?

Reviewer #1: Yes

Reviewer #2: Yes

5. Review Comments to the Author

Reviewer #1: Dear Editor and Authors:

Thank you for the opportunity to review this manuscript titled Expanding radiogenic strontium isotope baseline data for central Mexican paleomobility studies. The manuscript is well written and succinct. This research is aimed at producing more strontium isotope baseline data for future studies. It incorporates bioavailable strontium isotope data from previous studies and provides 63 new data point collected from plants and water samples. Although these types of studies do not provide new interpretations of past mobility per se, they are very important for future strontium isotope research in the region.

There is a good review of the background behind strontium isotope analysis, the geology of the region, the questions surrounding paleomobility in central Mexico. I would like to see the authors expand on the discussion of their sampling method, specifically how they chose the root length of the plants and what kind of information this would provide (i.e. top soil compared to deeper soil levels). Please detail how the edible parts of the plants were isolated for the current study. It would be interesting to present the difference between the edible plants and non-native/non-edible plants to see if these differences were significant.

The analytical methods that are applied work, but I feel like the data could be incorporated into strontium isoscape models that utilize global raster datasets in addition to the current baseline data. For examples of this type of research in the Caribbean and Western Europe, please see Bataille and Bowen 2012, Bataille, Laffoon and Bowen 2012, and Bataille et al. 2018.

Thank you

References:

Bataille CP and Bowen GJ. 2012. Mapping 87Sr/86Sr variations in bedrock and water for large scale provenance studies. Chemical Geology, 304: 39-52.

Bataille CP, Laffoon J and Bowen GJ. 2012. Mapping multiple source effects on the strontium isotopic signatures of ecosystems from the circum‐Caribbean region. Ecosphere, 3(12): 1-24.

Bataille CP, von Holstein IC, Laffoon JE, Willmes M, Liu XM and Davies GR. 2018. A bioavailable strontium isoscape for Western Europe: A machine learning approach. PLOS ONE, 13(5), p.e0197386.

Reviewer #2: Pachecho-Forés et al. review the state of bioavailable strontium values across central Mexico before providing their own baseline data in this study. They use cluster analysis to try to differentiate between sub-regions, but note that there is overlap in values between Basin of Mexico clusters.

I believe this paper fits as a PLOS ONE article and provided important data for the region. I provide the following notes and recommendations for revision.

The title and keywords are descriptive.

For authorship order, why is Pacheco-Forés listed as contributing equality to this work with no one else? Why is the symbol even necessary?

Line 30: few studies examine strontium variability within the Basin? Or none have?

Line 62: haven’t yet mentioned hydroxyapatite as what we analyze in tissues. Briefly explain.

Fig 1. Is this image derived from another image? Cite (I see it’s cited in-text, but cite in the caption). Label the 3 morphotectonic provinces in the figure.

Fig 2. Who made this map? You’ve jumpted ahead to the clustering which haven’t been introduced in the text yet. Confusing narrative. Also, why are most points off the map? Confusing.

Line 121. Define cluster membership.

Your methods section are arguably more detailed than necessary, but that’s fine.

Could you please provide your R code as Supplementary Information? See Styring et al. 2017 DOI: 10.1038/nplants.2017.76 for an example.

Line 233: ‘This rang is relatively large when…’ Well….yes. That’s not the point, you expected variation higher than analytical precision, you’ve already pointed that out in your intro. Delete this sentence.

Table 1. with this many samples, a supplementary table of raw data might be best and a summary table in its place as Table 1

Fix superscripts on y-axes of figures with strontium data.

Line 310: specify that oxygen is δ18O

References:

PLOS doesn’t give you a copy editor to prepare your manuscript for publication, ensure you’ve corrected all typos including superscript errors. References such as 27,28,29,32,38,40,42, and 54 need to be fixed.

References 52,53, and 54 are all conference papers. I would remove them.

Love that there’s a Spanish translation of the paper readily available!

Note to the editor: would a bilingual abstract and/or keywords be available at PLOS ONE, should the authors wish it?

6. PLOS authors have the option to publish the peer review history of their article (what does this mean?). If published, this will include your full peer review and any attached files.

Reviewer #1: No

Reviewer #2: Yes: Chris Stantis

---

## [Author Response · Author response to Decision Letter 0]

10 Feb 2020

Dear Dr. Halcrow,

We would first like to thank you and the reviewers for your helpful comments on our manuscript. We have carefully revised the manuscript based on reviewer comments, which we feel have greatly improved the clarity of our sampling methodology, as well as the presentation of our analytical results. We present our detailed responses to reviewer comments below. Thank you for your time and consideration.

Best,

Sofía I. Pacheco-Forés, Gwyneth W. Gordon, and Kelly J. Knudson

Academic Editor’s Comments

Please ensure that your manuscript meets PLOS ONE’S style requirements, including those for file naming.

SIPF: All files have been renamed in accordance to PLOS ONE editorial guidelines and we have carefully edited the manuscript to ensure it meets PLOS ONE’s style requirements.

In your Methods section, please provide additional location information, including geographic coordinates for the data set if available.

SIPF: These data are presented in Table 1 in the Results and Discussion section. We have also added an in-text reference to these data in S1 File in the Methods section to clarify that these data are available.

We note that Figures 1 and 2 in your submission contain map images which may be copyrighted […] Please check copyright information on all replacement figures and update the figure caption with source information. If applicable, please specify in the figure caption text when a figure is similar but not identical to the original image and is therefore for illustrative purposes only.

SIPF: The Figure 1 and 2 map images were made by SIPF using public domain vector map data from Natural Earth. We have cited morphotectonic province source data in the Fig 1 caption and have included Figure authorship and acknowledgements of Natural Earth vector maps in captions of both figures.

To enhance the reproducibility of your results, we recommend that if applicable you deposit your laboratory protocols in protocols.io, where a protocol can be assigned its own identifier (DOI) such that it can be cited independently in the future.

SIPF: Thank you for this suggestion! We are currently in the process of streamlining our laboratory protocols and will look into this option once our updates are finalized.

Reviewer One’s Comments

Thank you for the opportunity to review this manuscript titled Expanding radiogenic strontium isotope baseline data for central Mexican paleomobility studies. The manuscript is well written and succinct. This research is aimed at producing more strontium isotope baseline data for future studies. It incorporates bioavailable strontium isotope data from previous studies and provides 63 new data point collected from plants and water samples. Although these types of studies do not provide new interpretations of past mobility per se, they are very important for future strontium isotope research in the region.

SIPF: Thank you for your comments.

There is a good review of the background behind strontium isotope analysis, the geology of the region, the questions surrounding paleomobility in central Mexico.

SIPF: Thank you. We wanted to provide a good balance of strontium systematics, with more regionally specific geology and archaeological applications of paleomobility studies within Mexico.

I would like to see the authors expand on the discussion of their sampling method, specifically how they chose the root length of the plants and what kind of information this would provide (i.e. topsoil compared to deeper soil levels).

SIPF: This is an excellent point. We have added a description of our opportunistic sampling strategy and categorization of rooting depths (shallow root system <1 m deep, deep root system >1 m deep), as well as what kind of information plant rooting depth provides.

Please detail how the edible parts of the plants were isolated for the current study.

SIPF: We have further clarified that we isolated edible components of sampled plants by physically separating them from the dried collected plant.

It would be interesting to present the difference between edible plants and non-native/non-edible plants to see if these differences were significant.

SIPF: Thank you for this suggestion. We have incorporated a discussion of the differences between native edible plants simulating pre-Hispanic diets and non-edible/non-native plants that would not have contributed to pre-Hispanic human and animal strontium sources into the Results and Discussion. While the differences were not significant, we have included a S1 Fig as a figure further exploring these comparisons as a supplemental material in the Supporting Information.

The analytical methods that are applied work, but I feel that the data could be incorporated into strontium isoscape models that utilize global raster datasets in addition to the current baseline data. For examples of this type of research in the Caribbean and Western Europe, please see Bataille an Bowen 2012, Bataille, Laffoon and Bowen 2012, and Bataille et al. 2018.

SIPF: Thank you for this suggestion, as well as for the examples and references. We do see the ultimate goal of this project as incorporating these baseline data, along with others from the region into an isoscape model using global raster datasets. However, we are still working to compile and digitize regional geological maps and other relevant datasets for this purpose. In the meantime, we decided to prioritize publishing our generated Sr data with the current analytical methods for public use as we continue to work towards this goal.

Reviewer Two’s Comments

Pacheco-Forés et al. review the state of bioavailable strontium values across central Mexico before providing their own baseline data in this study. They use cluster analysis to try to differentiate between sub-regions but note that there is overlap in values between Basin of Mexico clusters. I believe this paper fits as a PLOS ONE article and provided important data for the region. I provide the following notes and recommendations for revision.

SIPF: Thank you for your comments.

The title and keywords are descriptive.

SIPF: Thank you.

For authorship order, why is Pacheco-Forés listed as contributing equally to this work with no one else? Why is the symbol even necessary?

SIPF: Symbols of equal authorship contribution have been removed.

Line 30: few studies examine strontium variability within the Basin? Or none have?

SIPF: This has been clarified.

Line 62: haven’t yet mentioned hydroxyapatite as what we analyze in tissues. Briefly explain.

SIPF: Thank you for catching this oversight. We included a brief explanation of the role of hydroxyapatite in the take-up and subsequent analysis of strontium in human and animal hard tissues.

Fig 1: Is this image derived from another image? Cite (I see it’s cited in-text but cite in the caption). Label the 3 morphotectonic provinces in the figure.

SIPF: We have clarified the authorship and source data of Fig 1 in the figure caption and have labeled the morphotectonic provinces in the figure, with a legend in the caption. Additionally, we have added sample site location information to Fig 1 to physically orient readers.

Fig 2: Who made this map? You’ve jumped ahead to the clustering which haven’t been introduced in the text yet. Confusing narrative. Also, why are most points off the map? Confusing.

SIPF: Thank you for your comment. We have added authorship and source data information in the figure caption. Additionally, we have re-worked this map to make it clearer. We moved Fig 2, which presents sampled central Mexican site cluster membership to the Results and Discussion section to improve the narrative flow of the paper. Additionally, we redrew the map so that the entire extent of central Mexico is visible. Within central Mexico, we highlight the Basin of Mexico, and include the symbology for both central Mexico and the Basin in the map legend.

Line 121: Define cluster membership.

SIPF: We have moved Fig 2 and its caption to the Results and Discussion section so that the reader is already familiar with the cluster analysis methodology when they encounter the figure.

Your methods section are arguably more detailed than necessary, but that’s fine.

SIPF: Thanks for your comment. We thought it best to provide more detail rather than less since we do not yet have our laboratory protocols published online with their own DOI.

Could you please provide your R code as Supplementary Information? See Styring et al. 2017 DOI: 10.1038/nplants.2017.76 for an example.

SIPF: Thank you for this excellent suggestion! We have provided both the data spreadsheet (S1 File) along with our R code in an annotated Markdown file (S2 File) in the Supporting Information section and cited them in-text.

Line 233: ‘This range is relatively large when…’ Well…yes. That’s not the point, you expected variation higher than analytical precision, you’ve already pointed that out in your intro. Delete this sentence.

SIPF: Deleted.

Table 1. With this many samples, a supplementary table of raw data might be best and a summary table in its place as Table 1.

SIPF: Thank you for this suggestion. While our preference is to include the raw data in the body of the paper for greater accessibility and transparency, we are happy to defer to the editor’s preference.

Fix superscripts on y-axes of figures with strontium data.

SIPF: Y-axis superscripts on Figures 3 and 4 have been fixed.

Line 310: specify that oxygen is �18O

SIPF: Corrected.

References: PLOS doesn’t give you a copy editor to prepare your manuscript for publication, ensure you’ve corrected all typos including superscript errors. References such as 27, 28, 29, 32, 38, 40, 42, and 54 need to be fixed.

SIPF: All superscript errors in References have been corrected.

References 52, 53, and 54 are all conference papers. I would remove them.

SIPF: These have been removed.

Love that there’s a Spanish translation of the paper readily available!

SIPF: Thank you for your comment. This is an important priority for us in encouraging accessibility and international collaboration. All of the above changes have been made in the Spanish translation supplementary manuscript as well.

Editorial Staff Comments

Thank you for stating in the manuscript Methods: 'All necessary permits were obtained for the described study, which complied with all relevant regulations. Samples were imported to the Arizona State University Archaeological Chemistry Laboratory under permits granted to Pacheco-Forés from the United States Department of Agriculture Animal and Plant Health Inspection Service (PCIP-17-00469).'

To comply with PLOS ONE submissions requirements for field studies, please provide the following information in the Methods section of the manuscript and in the “Ethics Statement” field of the submission form (via “Edit Submission”):

a) Provide the name of the authority who issued the permission for each location (for example, the authority responsible for a national park or other protected area of land or sea, the relevant regulatory body concerned with protection of wildlife, etc.). If the study was carried out on private land, please confirm that the owner of the land gave permission to conduct the study on this site.

b) For any locations/activities for which specific permission was not required, please

- State clearly that no specific permissions were required for these locations/activities, and provide details on why this is the case

- Confirm that the field studies did not involve endangered or protected species

SIPF: We have clarified in our Methods section that the Instituto Nacional de Antropología e Historia (INAH), the body governing the study sites, does not require specific permissions to collect water or modern plant samples from the study sites. Furthermore, no endangered or protected plant species were involved in the study. We have also included this information in our Ethics Statement field of the submission form.

Thanks again to reviewers for their comments!

---

## [Editor Report · Decision Letter 1]

12 Feb 2020

Expanding radiogenic strontium isotope baseline data for central Mexican paleomobility studies

PONE-D-19-30394R1

Dear Dr. Pacheco-Fores,

We are pleased to inform you that your manuscript has been judged scientifically suitable for publication and will be formally accepted for publication once it complies with all outstanding technical requirements.

With kind regards,

Siân E Halcrow, Ph.D.

Academic Editor

PLOS ONE
---

## [Editor Report · Acceptance letter]

14 Feb 2020

PONE-D-19-30394R1 

Expanding radiogenic strontium isotope baseline data for central Mexican paleomobility studies 

Dear Dr. Pacheco-Fores:

I am pleased to inform you that your manuscript has been deemed suitable for publication in PLOS ONE. Congratulations! Your manuscript is now with our production department. 

With kind regards,

on behalf of

Dr Siân E Halcrow 

Academic Editor

PLOS ONE